# Regulation of Intracellular Reactive Oxygen Species Levels after the Development of *Phallus rubrovolvatus* Rot Disease Due to *Trichoderma koningii* Mycoparasitism

**DOI:** 10.3390/jof9050525

**Published:** 2023-04-28

**Authors:** Meiling Lu, Tingchi Wen, Ming Guo, Qihua Li, Xingcan Peng, Yan Zhang, Zhenghua Lu, Jian Wang, Yanjun Xu, Chao Zhang

**Affiliations:** 1School of Pharmacy, Guizhou University, Guiyang 550025, China; 2State Key Laboratory Breeding Base of Green Pesticide and Agricultural Bioengineering, Key Laboratory of Green Pesticide and Agricultural Bioengineering, Ministry of Education, Guizhou University, Guiyang 550025, China; 3The Engineering Research Center of Southwest Bio-Pharmaceutical Resources, Ministry of Education, Guizhou University, Guiyang 550025, China; 4The Mushroom Research Centre, Guizhou University, Guiyang 550025, China; 5Guizhou Jinchandashan Biotechnology Co., Ltd., Nayong 553300, China; 6Guizhou Jinsun Biotechnology Co., Ltd., Zhijin 552100, China; 7Center of Excellence in Fungal Research, and School of Science, Mae Fah Luang University, Chiang Rai 57100, Thailand; 8The Key Laboratory of Agricultural Bioengineering, Guizhou University, Guiyang 550025, China

**Keywords:** edible fungi, *Trichoderma koningii*, mycoparasitism, ROS level, rot disease

## Abstract

*Phallus rubrovolvatus* is a unique mushroom used for medicinal and dietary purposes in China. In recent years, however, the rot disease of *P. rubrovolvatus* has seriously affected its yield and quality, becoming an economically important threat. In this study, samples of symptomatic tissues were collected, isolated, and identified from five major *P. rubrovolvatus* production regions in Guizhou Province, China. Based on combined analyses of phylogenies (ITS and EF1-*α*), morphological characteristics and Koch’s postulates, *Trichoderma koningiopsis* and *Trichoderma koningii* were identified as the pathogenic fungal species. Among these, *T. koningii* exhibited stronger pathogenicity than the other strains; thus, *T. koningii* was used as the test strain in the follow-up experiments. Upon co-culturing *T. koningii* with *P. rubrovolvatus*, the hyphae of the two species were intertwined, and the color of the *P. rubrovolvatus* hyphae changed from white to red. Moreover, *T. koningii* hyphae were wrapped around *P. rubrovolvatus* hyphae, leading to their shortening and convolution and ultimately inhibiting their growth due to wrinkling; *T. koningii* penetrated the entire basidiocarp tissue of *P. rubrovolvatus*, causing serious damage to the host basidiocarp cells. Further analyses revealed that *T. koningii* infection resulted in the swelling of basidiocarps and significantly enhanced the activity of defense-related enzymes, such as malondialdehyde, manganese peroxidase, and polyphenol oxidase. These findings offer theoretical support for further research on the infection mechanisms of pathogenic fungi and the prevention of diseases caused by them.

## 1. Introduction

*Phallus rubrovolvatus* (M. Zang, D. G. Ji and X. X. Liu) Kreisel typically grows beneath bamboo forests. Its basal fungal cord is connected to the bamboo whip and dead bamboo roots, and its delicate white mesh skirt is spread downward [1,2]. Owing to this very unique morphology, it is known as the “flower of fungi” or “empress of fungi” in Chinese [3,4]. Edible fungi rank as the fifth largest crop in China, playing an important role in the agricultural economy, with broad development prospects and immense research significance [5,6]. *Phallus rubrovolvatus* is a precious, large-scale, rare, medicinal, and edible mushroom in China, and its cultivation has paved a new way to overcome poverty among local farmers [3,7]. This mushroom is rich in polysaccharides and amino acids and produces beneficial effects such as anti-fungal activity, cancer prevention, lowering blood pressure, and weight loss, implying its broad development [8,9,10].

One hundred and nineteen *Phallus* species have been reported worldwide [11]. Among these, *P. rubrovolvatus* is popular and highly preferred by people. In China, the germplasm resources of *P. rubrovolvatus* are relatively rich. In particular, Zhijin County in Guizhou Province is popularly known as “the hometown of Chinese *P. rubrovolvatus*.” With Zhijin County as the center, the cultivation trend of this mushroom has radiated throughout Guizhou Province, becoming the most characteristic and advantageous edible fungus industry in this region [3]. *P. rubrovolvatus* prefers moderate temperatures and high moisture for growth. However, its unique warm and humid growth environment, characteristic odor, and rich nutrients are conducive to the occurrence of diseases and pests [12]. In addition, issues such as the availability of a single planting variety and strain degradation have resulted in an increasing incidence of diseases and pests, resulting in yield reductions of up to 60–70%, severely restricting the healthy development of the industry [7,13]. In addition, there are issues such as the availability of monoculture varieties and strain degradation

*Trichoderma* species, as representative biocontrol fungi, cause devastating damage to edible mushroom production, posing a serious threat to the development of this industry [14,15,16,17,18]. *Trichoderma oblongisporum*, for instance, has been reported to cause mycelial withering and death in *Lentinula edodes* due to its three to five times faster hyphal growth rate and ability to produce extracellular enzymes to attack the host [19,20]. In addition, *Trichoderma* species are more resilient to extreme temperature, humidity, and pH conditions and adversities [21]. Furthermore, *T. ganodermatigerum* and *T. koningiopsis* have been reported to cause green mold disease in *Ganoderma sichuanense* [22]. Moreover, *T. aggressivum* f. *aggressivum* is one of the major fungal pathogens of *Agaricus bisporus* [23]. However, little is known about the fungi that cause rot disease in *P. rubrovolvatus*. Recently, Chen et al. isolated *T. koningiopsis* of the *Trichoderma* species as a fungal pathogen causing green mold in *P. rubrovolvatus* [13].

To prevent pathogen invasion, the host activates its own defense system, which involves a series of responses [24]. For instance, phenylalanine ammonia-lyase (PAL) is one of the key enzymes involved in defense reactions [25]. The host generates an immune response to resist the invasion of pathogenic fungi. For instance, increased lignin production can strengthen the cell wall and increase tissue lignification, forming a mechanical barrier to pathogenic fungal invasion [26]. Ligninases include manganese peroxidase (MnP) and laccase. Laccase is a polycopper oxidase that can prevent the destruction of host cells by toxic compounds secreted by pathogenic fungi, and it has been reported to play a crucial defensive role in preventing pathogenic fungal infection in *Agaricus bisporus* [27]. Upon *Aspergillus niger* infection, basidiomycetes have been reported to enhance their defense response by augmenting lignin biosynthesis through a 67-fold increase in MnP activity and a 1.7-fold increase in laccase activity [28].

Despite extensive efforts in recent years to investigate, isolate, and control *P. rubrovolvatus* rot, the precise mechanisms of the pathogenic fungus’s invasion and spread into the host remain unknown; as a result, research on the interaction between *P. rubrovolvatus* and its pathogenic hyperparasite remains in its early stages.

In the present study, the pathogenic fungi causes of rot disease on *P. rubrovolvatus* in five planting regions in Guizhou Province, China, were isolated and identified. We aimed to isolate and identify the pathogenic fungi, evaluate their pathogenicity, and investigate the parasitism of pathogenic fungi on *P. rubrovolvatus*: (1) the morphological and ultrastructural characteristics of the infected basidiocarps were observed; (2) we measured important physiological and biochemical changes in the defense response; malondialdehyde (MDA), laccase, MnP, polyphenol oxidase (PPO), and PAL, were examined; (3) finally, the pathogenesis of rot disease and changes in the activities of the related defense enzymes were analyzed.

## 2. Materials and Methods

### 2.1. Field Investigation of Phallus rubrovolvatus Rot Disease

To determine the incidence of rot disease, samples were collected from five major production regions in Guizhou Province, China, including Nayong, Qianxi, Guiding County, Guiyang and Xingyi City. A hundred basidiocarps were chosen at random from each planting area. Three replicates were set up to count the number of rotting basidiocarps (counting when brown spots appeared), and the incidence rate was calculated as follows [29]:

Disease Incidence = [Number of infected plant units/Total number of plant units assessed] × 100

### 2.2. Isolation and Identification of Pathogenic Fungi and Evaluation of Their Pathogenicity

The samples of basidiocarps with rot disease were immersed in 0.1% HgCl for 3 min for disinfection and then washed with sterilized water three to five times. The tissues around the diseased spots were incised with a sterilized scalpel. Approximately a 2–3 mm^2^ portion at the junction of the diseased and healthy parts was sampled and inoculated on potato dextrose agar (PDA; 1 L potato infusion with 20 g·L^−1^ glucose and 15 g·L^−1^ agar); the plates were incubated at 25 °C for 3–5 days.

To test the pathogenicity of the isolated fungi, a sterile hole punch was used to sample a 5 mm^2^ fungal block. The fungal block was inoculated on the surface of healthy basidiocarps, six fungal basidiocarps were inoculated per strain, and three fungal blocks were placed per fungal basidiocarp. The samples were incubated in a smart mushroom fruiting chamber (LY-N7012, Zhucheng Liyu Machinery Co., Ltd., Sandong, China) at 25 °C and 80% humidity. Statistics were observed after inoculation, following the modified protocol of Tian et al. [30].

### 2.3. Species Identification and Confrontation Testing of Rot Disease Pathogenic Fungi

Morphological observation of the pathogenic fungus: The pathogens were inoculated on PDA medium and incubated at 25 °C for 3–5 days. The colony morphology, color, and growth rate of the pathogenic fungi were observed and recorded. Specifically, hyphae, conidia, and conidiophores of the pathogenic fungi were observed and recorded under a light microscope (DM2500, Leica, Weztlar, Germany), and their size was determined [31].

Molecular identification: DNA was extracted from fresh mycelium harvested from PDA plates after 4 days, as described by Turner et al. [32]. Fungal genomic DNA was extracted using the Fungal gDNA Isolation Kit (Biomiga, San Diego, CA, USA). With the primer pairs ITS5/ITS4 and EF1-728F/TEF1LLErev, the internal transcribed spacer (ITS), 5.8S ribosomal RNA, and partial translation elongation factor 1- (TEF) were amplified [33,34,35]. Polymerase chain reaction (PCR) was performed with a 25 µL reaction system containing 10 μL of PCR mix (Dream TaqTM Green PCR Master Mix 2×, Thermo, Waltham, MA, USA), 2 μL of template DNA, 1 µL each of forward and reward primers (10 µM), and 11 μL of ultrapure water. Amplification was performed on a T100^TM^ Thermal Cycler (BIO-RAD, Hercules, CA, USA), which was programmed for initial denaturation at 94 °C for 3 min, followed by 34 cycles of denaturation at 94 °C for 30 s, annealing at 51 °C for 50 s, extension at 72 °C for 45 s, and final extension at 72 °C for 10 min. The PCR products were sequenced using the same primers used for amplification at Sangon Biotech Co., Ltd. (Shanghai, China). The phylogenetic analyses were conducted using 61 strains, including 9 strains of *Trichoderma* species from this study; *Protocrea farinose* CBS 121551 and *P. pallida* CBS 299.78 were used as the outgroup taxa, and 50 other *Trichoderma* species selected sequences were obtained from the NCBI database (Appendix A).

Mycelium plate confrontation: When the mycelium of *Phallus rubrovolvatus* grew to 2–3 cm, *Trichoderma koningii* was inoculated at a distance of approximately 2 cm from the mycelium of *P. rubrovolvatus*. The plates were incubated in a constant-temperature incubator at 25 °C for confrontation culture.

### 2.4. Scanning Electron Microscopy (SEM)

The epidermis and inner layers of healthy and diseased basidiocarps were collected separately and placed for 2 h in vials containing 2.5% (*v*:*v*) glutaraldehyde at 4 °C. Thereafter, the samples were sent to the electron microscopy room for preparation. Briefly, the samples were washed three times with phosphate-buffered saline (PBS), post-fixed for 2 h in a 1% osmic acid solution, and washed with PBS again three times for 10 min. Then, the samples were dehydrated twice in an ethanol series (30%, 50%, 70%, and 95%, *v*:*v*) for 20 min, and finally in absolute ethanol. After drying, the samples were coated with gold films [36]. All samples were observed using FEI Inspect SEM (Hillsboro, OR, USA).

### 2.5. Transmission Electron Microscopy (TEM)

The sampling method was the same as that described in Section 2.4. The samples were placed in vials containing 3% (v) glutaraldehyde at 4 °C and then washed three times with PBS. Next, the samples were post-fixed for 2 h in a 1% osmic acid solution and washed with PBS again three times for 15 min. Then, the samples were dehydrated twice in an acetone series (30%, 50%, 70%, and 95%, *v*:*v*) for 20 min, and finally in absolute acetone. The dehydrated specimens were then polymerized for 48 h in 21-well silicate-embedded plates at 60 °C. Thin sections (thickness = 60 nm) were obtained with the Ultratome Leica UC6 and stained with 2% uranyl acetate (30 min) and lead citrate (10 min) [37]. The stained samples were then observed under a transmission electron microscope (JEM-1400FLASH, JEOL, Beijing, China).

### 2.6. Phallus rubrovolvatus Enzyme Activity Assays

When the fungus is stressed, it produces defensive enzymes to resist invasion. Therefore, the testing of defense enzymes is necessary. A sterile hole punch was used to sample a fungal block with a diameter of 5 mm; the fungal block was inoculated on the surface of healthy basidiocarps (3–5 cm). Tissues were sampled from disease spots and a 0.1 mm region around the disease spots after 3 days of operation according to the MDA and H_2_O_2_ test kit instructions (Beijing solarbio science and technology Co., Ltd., Beijing, China). The three types of samples can reflect the changes of defense enzymes under different health conditions. Then, 1 mL of MDA and H_2_O_2_ extract were added to 0.1 g of the sample. The samples were homogenized in an ice bath and centrifuged at 8000× *g* for 15 min at 4 °C on ice. The supernatant was used to determine MDA and H_2_O_2_ content [38].

PPO, MnP, PAL, superoxide dismutase (SOD), and laccase activity determination: The sampling method was the same as that for MDA quantification. Briefly, 1 mL of PPO, MnP, PAL, SOD, and laccase extracts were added to 0.1 g of sample. The samples were homogenized in an ice bath and centrifuged at 8000× *g* at 4 °C for 10 min [38,39,40,41]. The supernatant was placed on ice for testing. All test kits were purchased from Beijing solarbio science and technology Co. Ltd. (Beijing, China).

### 2.7. Tissue ROS Levels

Environmental stress induces the accumulation of reactive oxygen species (ROS) in the cells, which can cause severe oxidative damage to the plants, thus inhibiting growth and grain yield [42]. Pathogenic fungi were inoculated with healthy *P. rubrovolvatus* basidiocarps, and rot tissue was collected after 3 days. Intracellular ROS levels were measured using the In Situ Fluorescence Staining Kit (Genmed Scientifics Inc., Shanghai, China) according to the manufacturer’s protocol. The cells were observed under a confocal fluorescence microscope at the excitation and emission wavelengths of 499 and 515 nm, respectively. Intense green fluorescence indicates high ROS levels [43].

## 3. Results

### 3.1. Onset and Symptoms of Phallus rubrovolvatus Rot Disease

In the present study, the incidence of rot disease in five planting regions of Guizhou Province, China, was statistically analyzed, as shown in Table 1. The average incidence of greenhouse planting rot disease was the lowest in Guiding County and the highest in Qianxi County. The average incidence rate in greenhouse planting was 41.67–80.33%, and the average incidence rate of undergrowth planting rot disease was 6–36%.

Temperatures and humidity in the planting area during September were analyzed. (Table 2). Overall, temperature and humidity in the greenhouse were higher than those outdoors; therefore, warmer temperatures accelerated the reproduction of pathogenic microorganisms, resulting in a higher incidence of rot disease in greenhouse planting.

During the period around September, when a second crop of *P. rubrovolvatus* is grown and harvested, serious rot was recorded in both greenhouse and undergrowth planting. In particular, the incidence of rot disease is higher in greenhouse planting, with more severe symptoms. The disease primarily develops throughout the basidiocarp’s growth. At the initial stages of the disease, brown spots appear on the surface of *P. rubrovolvatus* basidiocarps, and their color gradually deepens, overflowing with light-yellow water droplets (Figure 1a,b). The disease spot gradually expands, forming obvious brown lesions, and gradually decays by exposing the glial layer tissue (Figure 1c), with white, yellow, or green colonies appearing around it. The diseased area expands rapidly (Figure 1d,e), resulting in the death or malformation of mushrooms after the extensive decay of the basidiocarps (Figure 1f,g,h).

### 3.2. Isolation of the Pathogenic Fungus and Evaluation of Its Pathogenicity

Six strains were isolated from the diseased basidiocarps of *P. rubrovolvatus*: NY120302, PL110114, PL218116D, PL218119D, XY101301, and NY120304H. We evaluated the pathogenicity of the isolated strains and found that PL110114 and NY120302 could cause *P. rubrovolvatus* rot, and both are pathogenic. Basidiocarps inoculated with NY120302 started presenting with reddish-brown spots on the surface after 2 days. The other test strains did not exhibit disease characteristics. Meanwhile, the basidiocarps inoculated with NY120302 and PL110114 started exhibiting spilled yellowish-brown water droplets on the epidermis after 3 days. The diameter of the spots gradually increased over time. On day 5, PL110114 damaged the basidiocarps more significantly, and the diseased area gradually expanded outward (Figure 2). At the later stages of the disease, the degree of decay increased, and the basidiocarps turned necrotic, with many white and green spore clusters attached. The process of symptoms was consistent with disease symptoms in the field. To verify that the same fungi caused the symptoms, the pathogens were isolated from the diseased basidiocarps and identified. Identified based on morphological and phylogenetic analyses, the identified pathogenic fungi were the same as the originally isolated strains. Therefore, PL110114 and NY120302 are the pathogens causing rot disease. PL110114 was collected from Baiyun District of Guiyang City, Guizhou Province, while NY120302 was collected from Nayong County of Bijie City, Guizhou Province.

### 3.3. Identification and Characterization of Fungal Isolates

All six fungi isolated from *P. rubrovolvatus* were identified as *Trichoderma* species. The non-pathogenic strains PL218116D, PL218119D, XY101301, and NY120304H were the most similar to and clustered with *Trichoderma pollinicola*, *Trichoderma lixii*, *Trichoderma tomentosum*, and *Trichoderma asperellum*, respectively. BLAST searches of the sequenced fragments revealed that the pathogenic strains NY120302 and PL110114 were the most similar to *T. koningiopsis* strain 18ASMA001 (ITS region: 100% identity to accession MT520621; EF1-*α*: 99.87% identity to accession MT671922) and *T. koningii* strain Hypo 51 = CBS 119500 (ITS region: 100% identity to accession FJ860762; EF1-*α*: 99.86% identity to accession KC285594), respectively. In addition, a phylogenetic tree was constructed with MEGA 7 based on the sequences of the ITS region and EF1-*α* gene to confirm that the representative isolates NY120302 and PL110114 shared high genetic similarity with *T.koningiopsis* and *T. koningii* (Figure 3). The highly pathogenic PL110114 was used in the follow-up experiments. Representative sequences of the tested DNA regions were deposited in the NCBI GenBank database (ITS: OP604608; EF1-*α*: OP620753).

The PL110114 strain grew rapidly on PDA medium when cultured at 25 °C for 4 days and overgrew the plates (9 mm). Aerial hyphae were white, and colonies were radial with neat edges (Figure 4); the conidia were initially white, which subsequently turned green. The conidiophores were multi-branched, opposite, or alternate. Phialides were solitary or 2–4 spirally arranged, flask-shaped, measuring 7–11 × 2–3 μm and 1.5–2 μm wide near the base. Conidia were sub-globose or obovoidal, measuring 3.0–4.2 × 2.5–3.6 μm.

### 3.4. Mycoparasitism Assays In Vitro

During the confrontation culture, the color of the *P. rubrovolvatus* mycelium changed from white to red when it came into contact with *T. koningii* (Figure 5a,b). Throughout the culture process, the color change of the hyphae implied aging and growth inhibition. Light microscopy revealed that after co-culture, *T. koningii* hyphae grew along the mycelium of *P. rubrovolvatus* and produced many sub-hyphae (Figure 5c); the hyphae grew close to the surface of the mycelium of *P. rubrovolvatus*, and their growth was significantly better than that of the *P. rubrovolvatus* mycelium (Figure 5d). The hyphae continued to wrap around the host in a spiral manner such that the mycelium of *P. rubrovolvatus* appeared shriveled (Figure 5e). The pathogenic mycelium grew profusely on top of the host hyphae to form a tumor-like protrusion, tightly wrapping the mycelium of *P. rubrovolvatus*. Eventually, the mycelium of *P. rubrovolvatus* shriveled and was unable to absorb nutrients from the medium, which led to growth inhibition or death (Figure 5f).

### 3.5. Morphological, Ultrastructural and Physiological Changes Process of Trichoderma koningii-Infested Phallus rubrovolvatus

#### 3.5.1. Morphological Studies by SEM

SEM revealed that healthy basidiocarp tissues were well arranged, smooth, full, and evenly distributed (Figure 6a,b). The dissection of basidiocarps after *T. koningii* inoculation revealed abundant invasive mycelium wrapped in the cells of infected *P. rubrovolvatus* basidiocarps; the infesting mycelium was thick and full and penetrated the intercellular spaces of *P. rubrovolvatus*. Meanwhile, the host mycelium was severely contracted and dried (Figure 6c,d).

#### 3.5.2. Ultrastructural Studies by TEM

TEM revealed that the cells of healthy basidiocarps were neatly arranged and evenly distributed (Figure 7a,d). In contrast, the cells of the basidiocarp infested with *T. koningii* were loosely arranged, and pathogenic cellular structures with dense cytoplasms and abundant organelles were observed in both the outer and inner cortical tissues. The tips of pathogenic mycelium contacting *P. rubrovolvatus* tissues formed tuberculate protrusions attacking the host, sinking the host cell wall inward (Figure 7b,c). The pathogenic mycelium tip gradually tapered to penetrate the interior of the cell, and the host cell structure was destroyed (Figure 7e,f).

#### 3.5.3. Effect on Defense Enzyme Activity

As shown in Figure 8, at 3 days after inoculation, the MDA content around and at the lesion site of basidiocarps was significantly higher than the CK (healthy tissue) value. As such, MDA content around and at the lesion site was, respectively, 18.41 and 90.66 times higher than that in the healthy areas of the basidiocarp, indicating the beginning of lipid oxidation around the rot and severe lipid oxidation at the rot site. PPO and PAL are important defense enzymes, while MnP and laccase play a defensive role in fungi [44]. As shown in Figure 8, PPO activity around and at the lesion site was, respectively, 36.35 and 51.03 times higher than that in the healthy areas of the basidiocarp. PAL activity around and at the lesion site was, respectively, 2.58 and 2.87 times higher than that in the healthy areas of the basidiocarp. Further, MnP activity around and at the lesion site was, respectively, 71.10 and 62.22 times higher than that in the healthy areas of the basidiocarp. Laccase activity around and at the lesion site was, respectively, 0.06 and 0.24 times higher than that in the healthy areas of the basidiocarp.

#### 3.5.4. Changes in ROS Levels

Compared with normal levels, ROS and H_2_O_2_ concentrations around and at the lesion site of *P. rubrovolvatus* tended to increase following *T. koningii* infestation, as evidenced by more-intense green fluorescence around and at the lesions. Under normal conditions, H_2_O_2_ levels in *P. rubrovolvatus* basidiocarps were 0.3 µmol·g^−1^ FW. Meanwhile, after *T. koningii* infestation, H_2_O_2_ accumulated around and at the lesion site at four-times-higher levels than normal, consistent with the results of green fluorescence (Figure 9). Under normal physiological metabolic conditions, ROS are continuously produced and scavenged simultaneously, and in the equilibrium state, ROS serve the function of maintaining the normal metabolism of the organism. In contrast, after *T. koningii* infestation, a reduction in ROS scavenging ability in the body of *P. rubrovolvatus* leads to an increase in their content, resulting in the oxidation of membrane lipids and oxidative damage to the cell membrane.

## 4. Discussion

*Phallus rubrovolvatus* rot is an important fungal disease commonly occurring on the second crop. The disease usually appears during the basidiocarp development period and is characterized by droplet-like dewdrops in the early stages. Subsequently, the epidermis starts rotting due to infection by various fungi, resulting in 60–70% disease incidence, and the basidiocarps deform or stop growing thereafter, leading to substantial economic losses for mushroom farmers [13,45,46].

*Phallus rubrovolvatus* rot is a typical soil-borne disease prone to outbreaks and devastating losses at high temperatures and humidity. It is a potentially harmful and important disease in the production of *P. rubrovolvatus. Trichoderma koningii* was identified as the pathogen causing *P*. *rubrovolvatus* rot. There are many species of pathogenic in edible fungal diseases, such as *Agaricus bisporus* disease, mainly *Pseudomonas tolaasii* and *Lecanicillium fungicola* [47,48]. Some studies have shown that the pathogens in *P. rubrovolvatus* are *Saccharomycopsis phalluae*, *Penicillium citrinum* and *T. koningiopsis* [3,13,49]. This work enriches the species of pathogens of *P. rubrovolvatus*.

This pathogenic fungus primarily affects the basidiocarps of *P. rubrovolvatus*, and the diseased part starts to rot at high temperatures and humidity; finally, green spores are visible. The mycelia of pathogens were white and separated. *Trichoderma pollinicola*, *T. lixii*, *T. tomentosum*, and *T. asperellum*, respectively, form a cluster on the phylogenetic tree. However, none of these four species was pathogenic. In contrast, strains NY120302 and PL110114, which lead to *P. rubrovolvatus* decay, were significantly pathogenic and most similar to *T. koningiopsis* and *T. koningii*, respectively (Figure 3). Among these, *T. koningiopsis* was identical to the isolated pathogenic fungi of green mold [13]. *Trichoderma* species contaminate the substrates of *Auricularia heimuer* and *Pholiota adipose*, infesting the mycelia of *P. pulmonarius,* which could inhibit fruiting body formation [50,51,52]. Furthermore, these pathogens could infest the fruiting bodies of *A. bisporus,* causing solid rot [53,54]. Overall, the morbidity symptoms described in previous studies were similar to the symptoms of *P. rubrovolvatus* rot disease observed in the present study.

Through disease surveys in the field, we noted that the incidence of rot disease in undergrowth planting was significantly lower than that in greenhouse planting, and the *P. rubrovolvatus* rot disease was more severe in the major planting areas of Guizhou Province, such as Qianxi, Nayong, and Xingyi. The disease typically occurs from July to September, coinciding with the growth and harvesting periods of a second crop of *P. rubrovolvatus*. Previously accumulated insect pests and pathogens, coupled with the high temperature and humidity conditions, reduce the immunity of mushrooms, rendering them susceptible to pathogenic fungi and resulting in serious decay. The main reasons for analyzing the prevalence of rot disease were as follows: (1) the amount of pathogenic fungal inocula accumulates over time; as the first crop accumulates many pests and diseases, the second crop is prone to a large outbreak [55,56]; (2) high temperatures and humidity are conducive to disease development [46]; (3) the soil in the study regions is sticky and heavy, poorly aerated, and prone to waterlogging, favoring disease development [57]; (4) fungi in soil and compost can cause a range of diseases in *P. rubrovolvatus* due to soil community imbalance, and *P. rubrovolvatus* mycelium can act as a substrate for these fungi [58]; and (5) ants, flies, mites and snails, among other carriers, harbor *Trichoderma* spores, acting as vectors for pathogen transmission [59,60]. The climate in Guizhou Province is warm and humid, with abundant rainfall, and the major cultivation areas of *P. rubrovolvatus* mostly constitute deciduous forests. During the summer, field humidity and temperature are high, which is favorable for the outbreak of *P. rubrovolvatus* rot disease [61]. The prolongation of *P. rubrovolvatus* cultivation may lead the pathogens to gather and multiply under the right conditions, and under the conditions of high temperature and humidity, this may result in a severe disease outbreak. In the present study, we confirmed the pathogenic fungus of *P. rubrovolvatus* rot; investigated the parasitism of pathogenic fungi on *P. rubrovolvatus*; observed the morphological and ultrastructural characteristics of the infected basidiocarps; and measured important physiological and biochemical changes in the defense response. In addition, we investigated the disease’s epidemiological factors. Our results can serve as an important guideline for the development of rot prevention and control measures during mushroom production.

Mycoparasitism is one of the important characteristics of *Trichoderma*. *Trichoderma* species are traditionally considered necrophilic re-parasitic fungi [62], playing important pathogenic roles in the host pathogenesis, primarily through entanglement, branching production, stinging, and the production of degradative enzymes to lyse the host mycelium [63,64,65]. To date, studies on the infestation of edible and medicinal fungi, such as *Pleurotus pulmonarius* and *Ganoderma* sp., by *Trichoderma* spp. have been conducted, while only a few studies have been reported on the infestation of *P. rubrovolvatus* by *T. koningii* [22,51]. In this study, we systematically studied the processes of *T. koningii* infestation of the host, formation of rot, mode of infestation, and effect on host cell destruction from histopathological aspects.

*Trichoderma koningii* mycelium growth and infestation rates are fast. When *Phallus rubrovolvatus* mycelium basidiocarps were inoculated with *T. koningii*, within 8 h, the pathogenic mycelium radially grew from the center of the block to the edges. Reddish-brown spots were visible at 48 h after inoculation, and yellowish mucilage appeared at 72 h. The mycelium of *T. koningii* could penetrate the entire tissues of *P. rubrovolvatus* during the infestation process. The pathogen fed on the decomposing tissues, allowing for the continuous multiplication of mycelium and conidia and producing a profuse fungal mass. TEM revealed that the tip of the mycelium could form a bulge, invading the interior of the cell after contact, and resulting in severe damage to the cell structure. Over time, the tissue became visibly spotted, began to rot and leach, and a large number of cells at the spot died [66]. The color of the spots deepened, the basidiocarp began to crumple or ooze more yellow water, the spots expanded and merged, and a large number of cells died, eventually leading to the cessation of growth or death of *P. rubrovolvatus* as it could no longer absorb nutrients normally [22,64,67].

After co-culturing on PDA plates, *P. rubrovolvatus* mycelium contacted the mycelium of *T. koningii*, and the reddening of the host mycelium could be visualized, followed by the cessation of growth. Microscopic observations revealed that the pathogenic mycelium intertwined with the host mycelium, producing numerous branching entanglements (Figure 5). *Trichoderma harzianum* infests *L. edodes* by forming mycelial entanglements at the anterior end of mycelia [67]. *Trichoderma atroviride* can penetrate the cell walls of other fungi upon interacting with them, grow inside their mycelium, and cause local cell death within 6 days of inoculation by breaking down the mycelial cell wall and feeding on the cytoplasm [68]. Thus, lesion formation is closely related to mycelial growth, which plays an important role in the pathogenesis process. In this study, after recognizing the host, *T. koningii* mycelium entangled with *P. rubrovolvatus* mycelium and produced many small branches with bulbous projections at the tips. From these observations, the invading mycelium likely produced a large amount of actin to form a network-like structure and ensure a strong tip shape, generating more mechanical pressure to squeeze the host cell wall inward to form the invasion point [69]. We hypothesize that this spherical structure is similar to the cell attachment structure that produces extracellular enzymes or toxins that cause damage to host tissue cells [70]. Here, we systematically studied the key infestation process of *T. koningii* on the host *P. rubrovolvatus* and clarified the pattern and structure of the infestation of this pathogen, as well as its destructive effect on host tissue cells.

Infestation by pathogens, such as fungi, bacteria, viruses, and nematodes, can increase ROS levels in the host, which are regulated via enzyme production through complex mechanisms of signaling pathways in response to external factors, and defense enzyme activity varies depending on the degree of damage to the host [71,72]. *Trichoderma* spp. produce an array of extracellular enzymes that ablate the host when invading it [73,74,75]. Krupke et al. showed that *Trichoderma aggressivum* f. *aggressivum* produces antimicrobial enzymes in mushroom compost that inhibit the mycelial growth of *A. bisporus* [76]. In Korea, oak wood mushrooms are primarily infested by *Trichoderma* species, which produce lignin-degrading extracellular enzymes that inhibit the host mycelial growth [77]. Upon infestation by *Trichoderma* spp., which cause green mold, it has been reported as a major constraint and common disease that leads to extensive damage to the mycelium and sporocarps of edible fungus [78,79]. When the pathogen invades the host, a series of defensive responses results in the upregulation of many lignin-like enzymes, such as laccase, MnP, and lignin peroxidase [27,80]. In our study, *P. rubrovolvatus* infested with *T. koningii* activated the key defense enzyme system in vivo, with a significant increase in the activity of defense enzymes such as PAL, SOD, and PPO; these results are consistent with most previous reports in other crops [15]. Hence, *T. koningii* infestation altered the activity of *P. rubrovolvatus* defense enzymes. Furthermore, upon infestation, excess ROS accumulate in the host body, leading to the peroxidation and de-lipidation of membrane lipids and damaging cellular structure and function. Finally, increased membrane permeability and the massive extravasation of intracellular solute molecules block the physiological metabolism of the host [44,80].

After recognizing the host, *Trichoderma* can inhibit its growth either by parasitism the living host to obtain nutrients or by secreting degradative enzymes that penetrate the cell wall and kill the host cells to obtain nutrients from the dead cytoplasm [81,82,83]. After the inoculation of *T. koningii* on *P. rubrovolvatus* basidiocarps, the mycelium grew rapidly and invaded the tissue. After invasion, the mycelium was closely attached to and entangled with the host hyphae. Microscopic observation revealed that *P. rubrovolvatus* mycelium appeared significantly swollen, and the mycelium and spores of *T. koningii* were observed inside the tissue. *T. koningii* mycelium may secrete extracellular enzymes upon contact with the host and then degrade and penetrate the cell wall of the host mycelium. TEM revealed that the pathogenic fungal mycelium could penetrate the host mycelial cells and enter the interior, causing serious damage. Furthermore, *T. koningii* infestation altered the activity of defense enzymes in *P. rubrovolvatus*. Specifically, the MDA content at and around the decay site increased significantly, and the activities of MnP, PPO, and other defense enzymes were greatly elevated, indicating that the invasion of the pathogenic fungus stimulated defense enzyme synthesis in *P. rubrovolvatus* to resist the attack. As the infestation time increased, the defense enzyme activity exceeded a certain level; simultaneously, the pathogenic fungus entered the cell and decomposed the cell wall of the host by producing extracellular enzymes, which led to the leakage of soluble sugars from *P. rubrovolvatus* cells. The degraded polysaccharides and proteins recruited other fungi or microorganisms to decompose or ferment the intracellular organic matter. The fermented organic matter inside the cells mixed with the cytosol to form droplets, leading to the appearance of brown spots. These spots gradually expand and eventually lead to basidiocarp rot.

Under the conditions of high temperature and humidity in the major cultivation areas of *P. rubrovolvatus*, *Trichoderma* species proliferate in the soil. Abundant *Trichoderma koningii* mycelium attaches to *P. rubrovolvatus* mycelium. *T. koningii* mycelium intertwines with *P. rubrovolvatus* mycelium through structures similar to cell attachments. *T. koningii* crushes the cell wall of *P. rubrovolvatus* inward by secreting cell wall-degrading enzymes and generating mechanical pressure through cell attachment-like structures to produce invasion sites. *T. koningii* invasion activates MAPK signaling in *P. rubrovolvatus* cells, inducing the expression of defense-related genes and the synthesis of defense enzymes, such as PPO, PAL, and MnP (Figure 10). *T. koningii* secretes extracellular enzymes, such as cell wall-degrading enzymes, which hydrolyze the cell wall components of *P. rubrovolvatus*, and the degraded polysaccharides and proteins attract flying insects or other microorganisms, further promoting the release of organic matter inside the cells. ROS accumulation exceeds the scavenging capacity of antioxidant enzymes (POD, SOD, and CAT), leading to membrane lipid peroxidation and oxidative damage, partial cell wall disintegration, organelle degradation, and cell content spillage in *P. rubrovolvatus*. The oozing cell sap mixes with the fermentation liquid to form droplets, leading to the appearance of brown spots, which gradually expand, eventually leading to basidiocarp rot.

In summary, the present study systematically described the process of spot formation after the infestation of *P. rubrovolvatus* by *T. koningii*, the mode of infestation, and the histopathological process of the destruction of host cells, providing a theoretical basis for the prevention and control of *P. rubrovolvatus* rot and the development of disease-resistant *P. rubrovolvatus* strains, promoting the production of edible mushrooms.

## 5. Conclusions

*Trichoderma koningii* can severely damage both the mycelium and basidiocarps of *P. rubrovolvatus*. Upon interacting with the *P. rubrovolvatus* mycelium, the *T. koningii* mycelium intertwines in a spiral manner, inhibiting the growth of and ultimately killing the host as it can no longer absorb nutrients normally. After infecting the basidiocarp, the pathogenic mycelium can penetrate the whole substrate. The tip of the pathogenic mycelium forms a tumor-like protrusion and invades the tissue, disrupting the physiological balance and defense response, which results in the malformation or rotting of *P. rubrovolvatus* basidiocarps. Our findings can serve as a useful reference for the prevention and management of *P. rubrovolvatus* rot disease.

## Figures and Tables

**Figure 1 jof-09-00525-f001:**
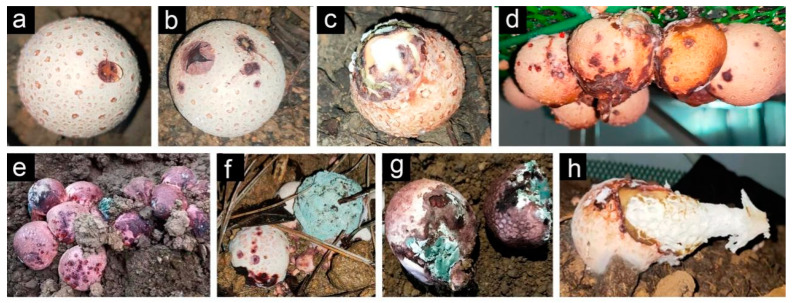
The pathological process of *P. rubrovolvatus* basidiocarp disease rot. (**a**) At the beginning of the disease, reddish-brown spots appear and overflow with water droplets. (**b**) The spots spread to the surrounding area. (**c**–**e**) In the middle and late stages of the disease, the spots expand or merge and infect a large area. (**f**,**g**) Whole basidiocarps rotten. (**h**) Formation of deformed mushrooms.

**Figure 2 jof-09-00525-f002:**
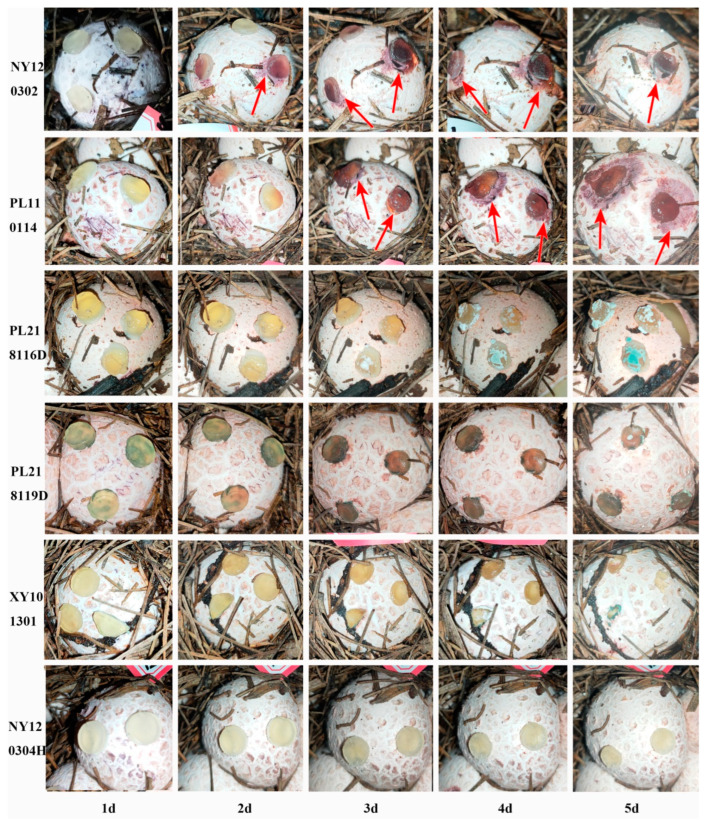
Pathogenicity identification. Inoculation of NY120302, PL110114, PL218116D, PL218119D, XY101301, NY120304H strains (**1d**–**5d**).

**Figure 3 jof-09-00525-f003:**
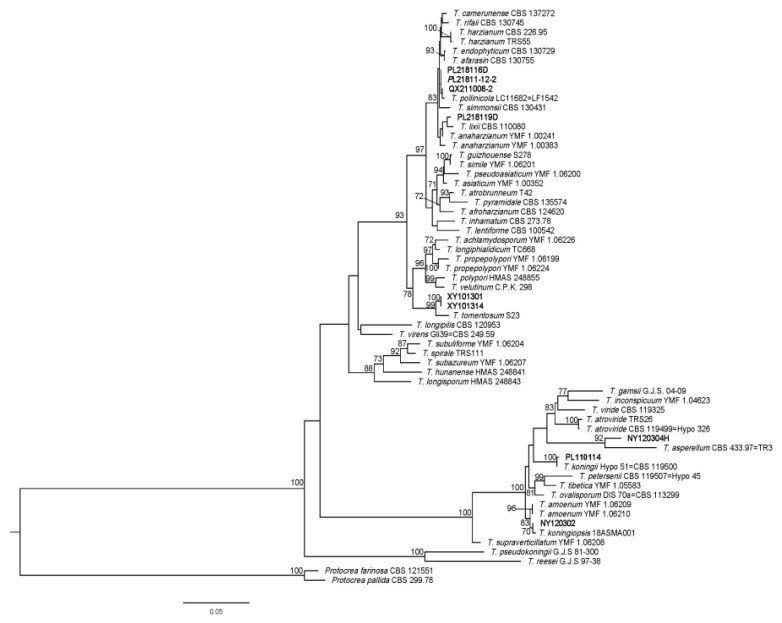
Phylogram generated from maximum likelihood analysis of ITS and EF1-*α* sequence data. Bootstrap support values for maximum likelihood; greater than 70% is given above branches. The tree is rooted with *Protocrea farinosa* CBS 121551 and *P. pallida* CBS 299.78. This study species are indicated in black bold.

**Figure 4 jof-09-00525-f004:**
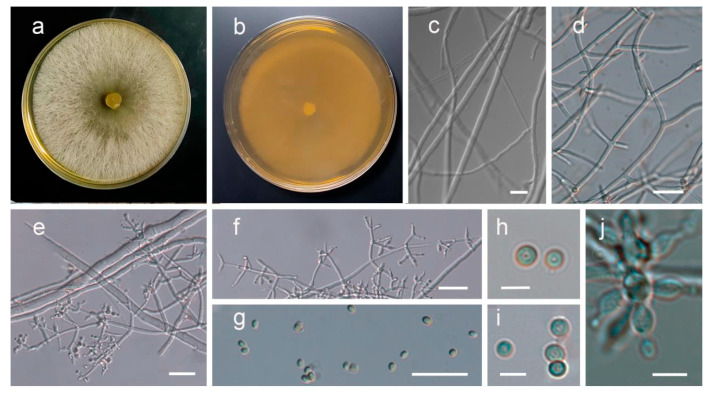
Morphological characterization of *T. koningii*. (**a**,**b**) Front and back sides grown on PDA plates for 4 days. (**c**,**d**) Mycelium. (**e**,**f**) Conidiophores and phialides. (**g**–**j**) Conidia. Scale bars, 25 µm(**c**–**g**), Scale bars, 5 µm (**h**–**j**).

**Figure 5 jof-09-00525-f005:**
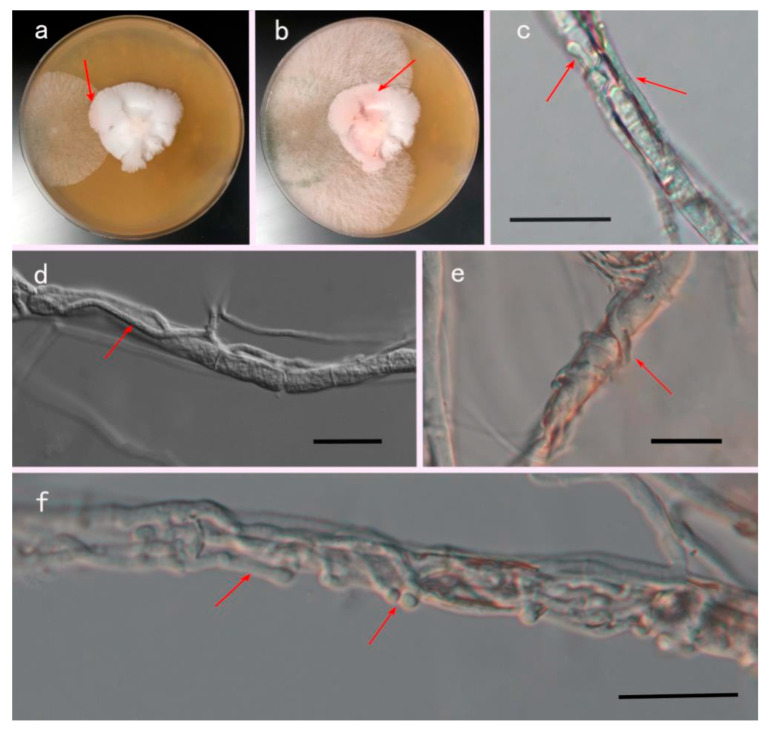
*T. koningii* confronts the mycelium plate of *P. rubrovolvatus*. (**a**,**b**) The hyphae of *P. rubrovolvatus* turned red when exposed to *T. koningii* hyphae. (**c**,**d**) *T. koningii* hyphae climbing on the hyphae of *P. rubrovolvatus*. (**e**) *T. koningii* hyphae mycelia entwined with *P. rubrovolvatus* hyphae. (**f**) *T. koningii* hyphae grow more small hyphae. Scale bars, 25 µm, the red arrow is the antagonistic position.

**Figure 6 jof-09-00525-f006:**
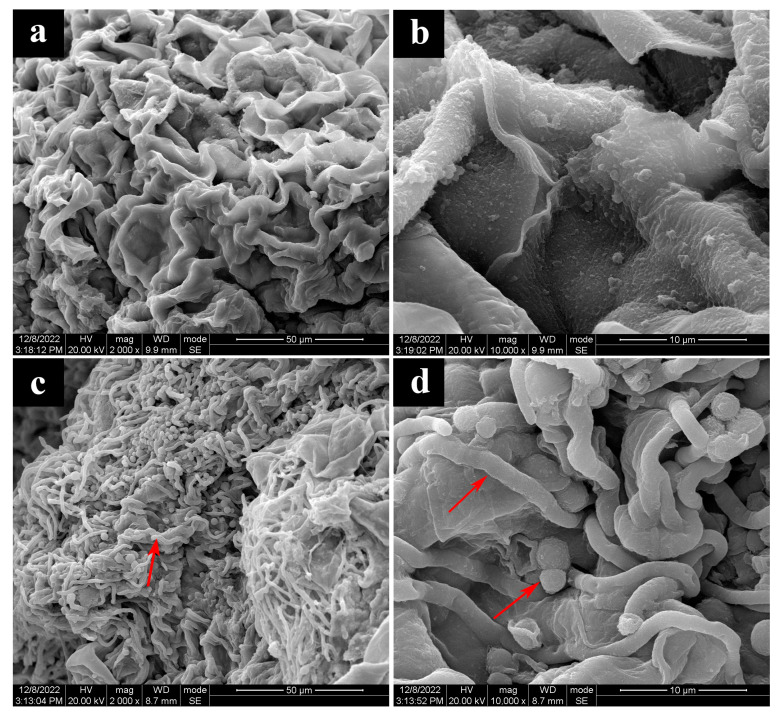
Morphological characteristics of the inner substratum tissues of basidiocarps of *P. rubrovolvatus*. (**a**,**b**) healthy basidiocarps. (**c**,**d**) rotten basidiocarps. Scale bars, 50 µm (**a**,**c**), Scale bars, 10 µm (**b**,**d**).

**Figure 7 jof-09-00525-f007:**
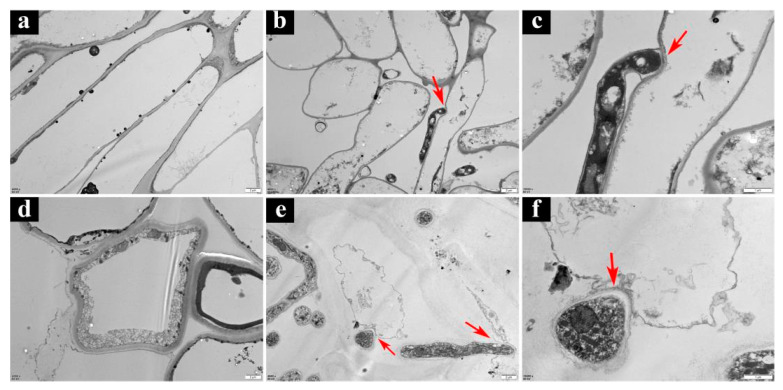
Ultrastructure of *P. rubrovolvatus* infected by *T. koningii*. (**a**) Epidermal layer cells of healthy basidiocarps. (**b**,**c**) Morphology of epidermal cells after infestation. (**d**) Inner layer cells of healthy basidiocarps. (**e**,**f**) Morphology of inner layer cells. Scale bars, 2 µm (**a**,**b**,**d**,**e**), Scale bars, 1 µm (**c**,**f**).

**Figure 8 jof-09-00525-f008:**
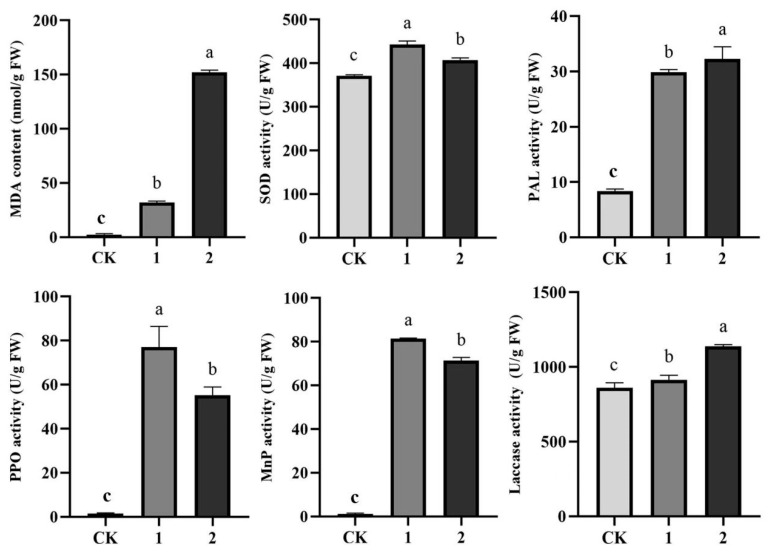
Effect of *T. koningii* infestation on MDA, MnP, PPO, PAL, laccase, and SOD activities. (CK) Tissue of healthy basidiocarps. (1) Disease spots. (2) 0.1 mm region around the spots. Different letters indicate a statistically significant difference (*p* < 0.05).

**Figure 9 jof-09-00525-f009:**
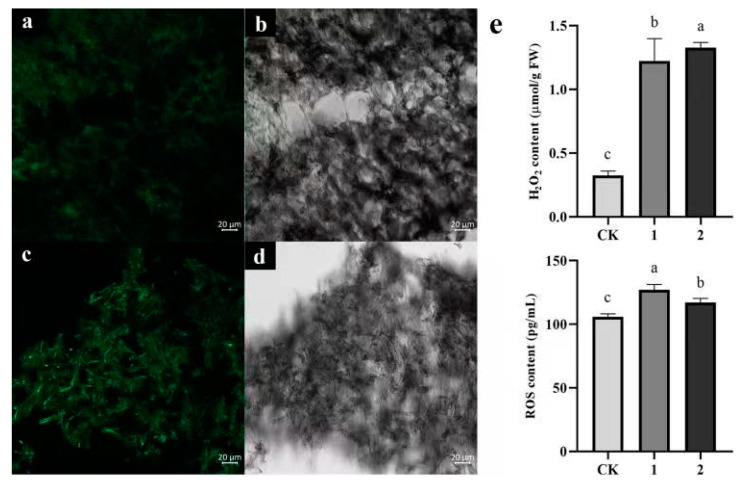
The effect of *T. koningii* infestation on *P. rubrovolvatus* reactive oxygen species. (**a**,**b**) Tissue from healthy basidiocarps. (**c**,**d**) Tissue after infection. (CK) Tissue of healthy basidiocarps. (1) Disease spots. (2) 0.1 mm region around the spots. Scale bars, 20 µm (**a**–**d**). (**e**) Statistically significant difference (*p* < 0.05) is indicated by different letters.

**Figure 10 jof-09-00525-f010:**
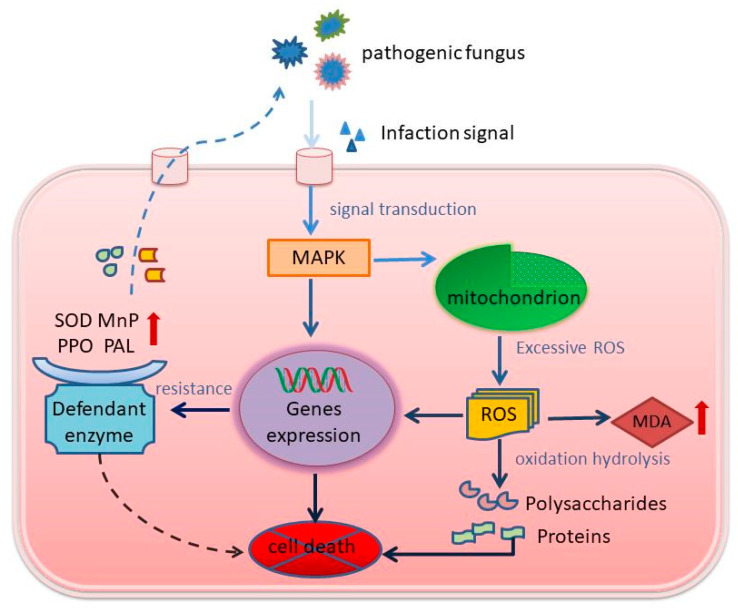
Pathogenic mechanism of *T. koningii* in *P. rubrovolvatus*.

**Table 1 jof-09-00525-t001:** Field incidence of *P. rubrovolvatus* rot disease.

	Greenhouse Planting	Undergrowth Planting
Growing Area	Average Incidence (%)	
Xingyi	66.33 ± 4.16	15.33 ± 9.02
Nayong	69.00 ± 14.42	36.00 ± 12.49
Qianxi	80.33 ± 12.22	36.00 ± 4.00
Guiyang	67.67 ± 3.06	6.00 ± 2.00
Guiding	41.67 ± 4.16	10.00 ± 2.00

Date: September 2021, Total statistics of basidiocarps in Greenhouse planting: *n* = 100 (pcs), Total statistics of basidiocarps in Undergrowth planting: *n* = 50 (pcs).

**Table 2 jof-09-00525-t002:** Temperature and humidity statistics in September in main *P. rubrovolvatus* growing areas.

Undergrowth Planting					Greenhouse Planting	
Growing Area	Mean Minimum Temperature/°C	Mean Maximum Temperature/°C	Average Monthly Temperature/°C	Average Humidity %	Average Monthly Temperature/°C	Average Humidity %
Qianxi	17.73	27.60	22.67	78.70	29.50	85.20
Nayong	17.70	26.83	22.27	80.50	28.30	90.30
Xingyi	18.57	28.47	23.52	84.43	30.20	95.00
Guiding	18.30	28.37	23.33	84.70	29.80	90.00
Guiyang	18.17	27.40	22.78	79.73	28.50	85.40

## Data Availability

Not applicable.

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
