# Peer review of "Regulation of Intracellular Reactive Oxygen Species Levels after the Development of Phallus rubrovolvatus Rot Disease Due to Trichoderma koningii Mycoparasitism"

_jof, 2023, doi:10.3390/jof9050525_

Round 1

Reviewer 1 Report

Dear authors,

This is an interesting study on Trichoderma hyperparasitism. Experiments conducted in this study were well carried out.The photoplates also appear excellent, the phylogentic tree is ok.

However, the paper needs substantial English language polishing, though! It needs to be edited by a mycologist! Many sentences are utterly unintelligible. It's a shame because the paper is excellent overall.

Another problem is that the writers mixed up the manuscript's results, discussion, and conclusion sections. The results section included some of the discussion and conclusions. 

Specific comments:

Abstract- lines 22-23- Samples of what? Did you collect samples of symptomatic tissues? It is not clear what did you collect.

Lines 24-25 And how did you identify these fungi as pathogenic? The phylogeny will not tell if a fungus is pathogenic. It is not clear if you conducted a pathogenicity test or not.

Introduction, line 91- What do you mean by types of fungi?

Line 96- Preliminary analyzed? Why preliminary?

Line 109-what is a tissue separation method? never heard of it

Line 118-you mean smart mushroom fruiting chamber? What is the producer?

Lines 119-120- Statistics were observed after inoculation to observe the pathogenesis, following a modified protocol of Tian et al” I don’t understand this sentence. Please, modify it to be understandable.

Line 124- fungi?? you mean fungus? Which light microscope? Which camera? How many conidia you count? Why did you measure the size of hyphae? How did you record a growth rate? On which temperatures? For how many days? Where is the experimental design for this?

Line 131-What is a reaction system?

Line 169-“directly covering the surface of healthy basidiocarps” I couldn’t understand this

Line 184- “Pathogenic fungi were inoculated with healthy P. rubrovolvatus basidiocarps, and plaque tissue was collected after disease onset” I couldn’t understand this sentence. And what is a plaque tissue??

Line 191- Onset?

Line 199- “Hence, the causative fungus of this disease must be identified, and the development of disease prevention and control strategies is imperative.” This sentence does not belong to the Results part of the manuscript!

Table 2- Again what is an onset investigation?

Lines 210-214 “Overall, temperature and humidity in the greenhouse were higher than those outdoors; therefore, warmer temperatures accelerated the reproduction of pathogenic microorganisms, resulting in a higher incidence of rot disease in the greenhouse planting. Hence, greenhouse planting may promote the occurrence of P. rubrovolvatus rot disease compared with undergrowth planting.” These sentences do not belong to the Results part of the manuscript!

Line 218-“serious rot is recorded in both greenhouse and undergrowth planting” I would use a past tense to describe the observed symptoms

Line 236- What did you mean by significantly pathogenic?

Line 245-Spore development?

Line 247-and how did you identify them?

Line 248- change to “…are the pathogens causing rot disease…”

Line 255- Change the title to “Molecular phylogentic identification”

Line 286-you mean strain confrontation experiment?

Line 289- “these findings are very interesting and warrant further attention” this is not part of a Results section

Figure 5- f. grow more small branches?

Line 314-“Therefore, T. koningii successfully penetrated the entire basidiocarp  tissue of P. rubrovolvatus, causing serious damage to the host basidiocarp cells.” this sonds alike a conclusion

Line 351- “Therefore, T. koningii infestation enhanced the activity of defense enzymes in host basidiocarps. In particular, the activity of three defense enzymes, namely PPO, PAL, and MnP, was significantly augmented.” Again a conclusion in the results part of the manuscript

Figure 10-Infaction signal?

Line 397-On which crop?

Lines 397-407- This whole paragraph should be Introduction! It has nothing to do with the discussion

Line 411 “with the accumulation of mycorrhizal sources”?what does this mean??

Line 413- “healthy development of the edible mushroom industry”?? This has no sense!

Line 417-Edible fungal diseases?? “many species of pathogenic in edible fungal diseases”??

Line 420-“This work enriches the species of pathogenic of P. rubrovolvatus and provides new 420 thoughts for control.” This is not understandable!

Line 427- These species were not pathogenic!

Line 431-What is significantly pathogenic?

Line 433- “Among these, T. koningiopsis was identical to the pathogenic bacterium of green mold isolated by…” Pathogenic bacterium of green mold??

Lines 434 “Trichoderma species contaminate the substrates of Auricularia heimuer and Pholiota adipose, infesting the mycelia of P. pulmonarius could inhibit fruiting body formation “ This has no meaning!

Line 437-Morbidity symptoms?

Line 443 “Previously accumulated insect pests and pathogens, coupled with the high temperature and humidity conditions, reduce the immunity of mushrooms, rendering them susceptible to pathogenic fungi and resulting in serious decay.” This sentence is not understandable!

Line 459- Accumulation of pathogens? what is accumulation of pathogens?

Line 469-Re-parasitic?

Line 557-Spot formation?

Author Response

请看附件

Reviewer 2 Report

Dear Authors,

The topic of paper titled “Regulation of intracellular reactive oxygen species levels after the development of Phallus rubrovolvatus rot disease due to Trichoderma koningii hyperparasitism” is interesting and contributes to the development and production of the important fungi  in the country's economy.

In general, the objectives must be improved. The structure of M&M and Results can be re-organized. Table 2 should become a figure including the standard deviation analysis.

Infection is the correct term and not infestation. With respect to Trichoderma, the term re-parasitism or hyperparasitism would not be appropriate, and it is convenient to speak of mycoparasitism. I understand that the proposed infection mechanism in Results corresponds to discussion.

All corrections, suggestions and comments are in the attached PDF.

L 59-60 rewrite

L 62. I don´t understand

L 64. rewrite

L 76. The year is missing.

L 90 rewrite "fungi causes of rot disease on Phallus ....

L92. Write with most precision the aims. 

You studied:

- Mycoparasitism of pathogen fungi against P. rub... in vitro

-Morphological and ultraestructural characteristics of infected basidiocarps

-biochemical /physiological changes....

L 92. replace determined by identified

L 105. replace plant by basidiocarps

L107. remove identification that goes in the other section

L 113. were taken and placed

L 116. Complete:

fungal mycelia discs from young culture of ....days???.

L 117. Separate with point. For this, ...

L 117-118. First, you may put the number of inoculated discs/blocks (repetitions) by basidiocarp, and then the number of replicates by fungal strain.

L 119. Change patogenesis by "develop of rot

L 121. I think that the topic is "mycoparasitism assays"

L 121. Fungal

L 122. Phenotypic characteristics

L 126. Change to: For molecular ....., DNA...

L143. re-arrange the structure of these sectiones and tittle

Ex.:

2.4. Study of hiperparasitism in vitro

2.5. Morphological and ultraestructural  characterisation

2.5.1. SEM

2.5.2.TEM

2.6 Physiological and biochemical characterisation

2.6.1 MDA and...

L 143. Introduce the assay as:

"In-plate confrontation assays, the presence of active hyperparasitism by Trichoderma koningii against P. rubrovolvatus was verified".

L. 144. Why do you use an isolate of T. koningii?

Briefly, describe. The most pathogenic isolate was used in the assay.

L 147. What are the objectives of these studies 2.4 and 2.5?

Modify the title and write an introductory sentence

L 168. Newly, an introductory sentence to the assay is missing

Briefly explain why you taken samples of threes types...

Maybe 2.6 and 2.7 can joined in the same section

L 183. Introduce briefly the objective of this assay.

L 184. How many isolates?

L 185. I don't understand

L.196. Move to Discussion and conclusions

L 202. I think it would be better to make a comparative graph by locality (greenhouse-undergrowth planting) only with the incidence averages and add the deviations. At the bottom of the figure, next to the title, place the n= of basidiocarps evaluated in each case.

L 202. Title: change "onset investigation" for "incidence"

L 205. This is repeated with the table. Delete, please!

L 2013. Move to Discussion

L252. Move the figure 2 to M&M

L 256. Change the title by Identification and characterisation of fungal isolates

L 256. Introduce the sentence. "All six fungi isolated from P. rubrovolvatus  were identified as Trichoderma species".

L 256. Move this sentence to M&M

L 286.

Add the title

Mycoparasitism assays in vitro

L. 289. This belong to  discussion

L. 305. Rewrite the title: Morphological, ultraestructural  and physiological changes....

L 319. roted basidiocarps

L 321. Modify. Ultraestructural studies by TEM

L 332. P. rubrovolvatus infected by T. koningii

L 337. This sentence corresponds to Discussion

L 340. Add: (healthy tissue)

L. 345. ??

L 375. I think that this section should go up for discussion

L 420. rewrite

L 461. rewrite

L. 468. Mycoparasitism????

L. 515. ???

L 534. Why? "re"  Trichoderma is parasite or mycoparasite

L 556/557. Here, you can put the section 3.5
